# Simulation and Optimization of Supply and Demand Pattern of Multiobjective Ecosystem Services—A Case Study of the Beijing-Tianjin-Hebei Region

**Aibin Wu** [1,2]**, Jianwen Zhang** [2]**, Yanxia Zhao** [2]**, Huitao Shen** [2] **and Xiaoping Guo** [1,*]

[1] School of Soil and Water Conservation, Beijing Forestry University, Beijing 100083, China; wuaibin@bjfu.edu.cn

[2] Hebei Engineering Research Center for Geographic Information Application, Institute of Geographical Sciences Hebei Academy of Sciences, Shijiazhuang 050011, China; zjianwen2021@163.com (J.Z.); zhaoyx8698@163.com (Y.Z.); shenhuitao80@126.com (H.S.)

[*] Correspondence: guoxp@bjfu.edu.cn

**Abstract:** Assessing and predicting the impact of land use/cover changes on ecosystem service supply and demand are crucial to formulating effective sustainable land use policies. In this study, we use the ecosystem service (ES) score matrix, ES supply rate, and ES supply/demand ratio to analyze the supply/demand pattern of ecosystem services based on land use/cover changes in the Beijing-Tianjin-Hebei region from 1990 to 2020. The Conversion of Land Use and Its Effects-Simulation (CLUE-S) model is used to simulate the spatiotemporal patterns of land use change in three scenarios of natural development, ecological priority development, and economic priority development and to predict and simulate the evolution of the ES supply and demand patterns in these different scenarios from 2030 to 2050. It was found that the main land use types are farmland and woodland in the Beijing-Tianjin-Hebei region, accounting for more than 67% from 1990 to 2020, the proportion of farmland decreased from 51.79% to 46.11%, and the proportion of woodland increased from 20.99% to 21.34%; the land use transformation was mainly from farmland to construction land from 1990 to 2020. The supply of ES in the Beijing-Tianjin-Hebei region was at a high level, the supply rate of ES increased from 0.78 to 0.81, the supply/demand ratio of ES decreased from 0.33 to 0.16 from 1990 to 2020, and the supply and demand of ES in the northern and western parts of the Beijing-Tianjin-Hebei region were in surplus. In the natural development scenario, the ES in the Beijing-Tianjin-Hebei region would remain in a high supply state from 2030 to 2050, but the pressure would be greater than before. The deficit, centered on urban construction, would widen, and the ecological situation would deteriorate. In the ecological priority development scenario, the pressure on the ES would be relieved, and the rate of deficit expansion would be reduced. In the economic development priority scenario, the pressure on the ES would increase sharply, and the deficit area would expand rapidly.

**Keywords:** ecosystem services; ecosystem service matrix; CLUE-S; ecosystem services supply and demand; Beijing-Tianjin-Hebei region



## 1. Introduction

Ecosystem services (ES) are defined as the benefits people derive from ecosystems [1]. ES supply refers to the services that the ecosystem can provide for human society [2]. ES demand are the services that humans obtain from the ecosystem for survival and development [3]. The research on supply and demand of ecosystem services is an important foundation for optimizing regional ecosystem structure, and has important practical significance for realizing sustainable economic and social development [4]. Land use/land cover (LULC) is one of the important influencing factors of ES as the changes have complex impacts on ecosystem patterns and processes for ES assessment [5]. Regional land use change can significantly change the ecosystem patterns and processes, resulting in changes

of ecosystem services supplies [6]. LULC is a direct consequence of human and nature interactions [7], the focus of research includes the global scale [1,8], national scale [9], regional scale [10–12], metropolitan areas [13] and ecologically fragile areas such as wetlands [14], river basins [15] and mining areas [16]. The Beijing-Tianjin-Hebei region is the political and economic center of China and is one of the most developed regions in China. For this specific area, some studies analyzed the eco-environmental effects [17], ecological security pattern construction [18], carbon sequestration [19], and ecosystem services value [20] based on LULC, research on the relationship between LULC and ES supply and demand is relatively lacking. Thus, it is necessary to explore the comprehensive impact of human activities on ES supply and demand from the land use change perspective in the Beijing-Tianjin-Hebei region.

To date, the primary methods to assess the supply and demand of ES are the Ecosystem Supply and demand index method [21], the ecological model method [22,23], valuation method and participation method [24,25]. Burkhard first proposed the expert assessment matrix method for ES [26] and applied it to assess the supply and demand of irrigated rice system services in Vietnam and the Philippines. Burkhard suggested that increasing the number of respondents and adopting data integration methods could improve the accuracy of results [27]. This method is easy to operate and has low data requirements. It calculates regional ES through expert scores (0 to 5 points) representing matrix service supply and demand, so as to facilitate the comparison of the supply and demand of ES between regions. Based on the ES matrix approach, Md Shawkat Islam Sohel addressed the biophysical assessment of ES supply capacities of various land use/land cover (LULC) types in the Lawachara National Park and its surrounding areas in north-east Bangladesh [28]. Weyland used an expert assessment matrix method to assess the relationship between ecosystem functions and services in four different eco regions of southern South America [29]. Depellegrin used the ES matrix approach for the assessment of ES potentials and ES interactions over Lithuania's national territory [30]. Wangai applied the ES matrix approach to spatially display the potential for regulating ES in a data-scarce peri-urban region in Kenya [31]. In China, some research improved the expert scoring matrix according [32,33]. This study of the supply and demand of ES included the Lake River basin, the Beijing-Tianjin-Hebei region, and the Yangtze River Delta region. The expert scoring matrix method for ES is mainly based on LULC, which requires less data, is easy to operate, and is suitable for research at various scales.

Research on the supply and demand of regional ES is based on the existing land use data. How will the supply and demand pattern change in the future? To date, research on this problem is still rare. However, the related research results of land use change simulation and prediction are rich. Several simulation models are widely used, such as CA(cellular automation), the Markov model [34,35], LUCAS (land use and carbon scenario simulator) [36], SLEUTH (Slope, Land use, Excluded, Urban, Transportation, Hill shade) [37], ANN-CA (artificial neural network-cellular automata) [38], Geomod [39], and CLUE-S (Conversion of Land Use and its Effects at Small Region extent) [40], among which CLUE-S does not just integrate the regional natural and human elements, and improve the future land use/land cover change simulation research, but at the same time it can also comprehensively simulate the temporal and spatial changes of multiple land use types in different scenarios, providing a more scientific basis for land use decision-making. The CLUE-S model was invented by the Dutch Wach University of Ningen (formerly Wageningen Agricultural University). Studies by many scholars showed that the CLUE-S model has become an effective scientific tool for simulating changes in land use patterns in the future. Verburg used the CLUE-S model to analyze natural causes in the Philippines and Malaysia [40]. In the CLUE-S modeling framework, Batisania parameterized and validated the model based on land use data interpreted from Landsat TM images of Centre County in 1993 and 2000 to predict the size of metropolitan areas in the United States [41]. Trisura coupled the CLUE-S model and Evaluation model framework (GLOBIO) to assess biodiversity integrity, simulate land use patterns in northern Thailand in 2050, and analyze

the impact of various development directions on biodiversity [42]. Therefore, we can use the CLUE-S model to study the simulation and prediction of land use change in different scenarios, and on this basis, use the ecosystem service matrix to analyze the supply and demand pattern of ES based on future land use changes in different scenarios.

The Beijing-Tianjin-Hebei region is one of the most developed regions in China. The rapid expansion of its urban agglomerations has to a certain extent caused damage to the ecological environment. At present, the contradiction between the two has become particularly acute. With the growth of the population in Beijing and the development of heavy industry in the Tianjin and Hebei areas, severe ecological and environmental problems such as land subsidence, desertification, and sandstorms are becoming more and more serious [21]. In this study, we analyzed the land use change patterns in the Beijing-Tianjin-Hebei region from 1990 to 2020; then, we used the Conversion of Land Use and Its Effects-Simulation (CLUE-S) model to simulate future land use patterns in the Beijing-Tianjin-Hebei region in multiple scenarios. The change in the ecosystem service (ES) supply and demand pattern from 1990 to 2050 was studied using ArcGIS and the expert assessment matrix. The results of this study can provide scientific data support for regional development.

## 2. Materials and Methods

### 2.1. Study Area

The Beijing-Tianjin-Hebei region is located in the heart of the Bohai Rim (36°01′–42°37′N, 113°04′–119°53′E), including the Beijing and Tianjin municipalities and Hebei Province (Figure 1), with the Yanshan Mountains in the north, North China Plain in the south, Taihang Mountains in the west and Bohai Bay in the east. There are various landforms in the Beijing-Tianjin-Hebei region, with mountains, plateaus, and basins in the west and north, and plains in the east and south. The topography is high in the northwest and low in the southeast, with a total area of 21.66 × $10^4$ km$^2$, accounting for 2.27% of the total national area.

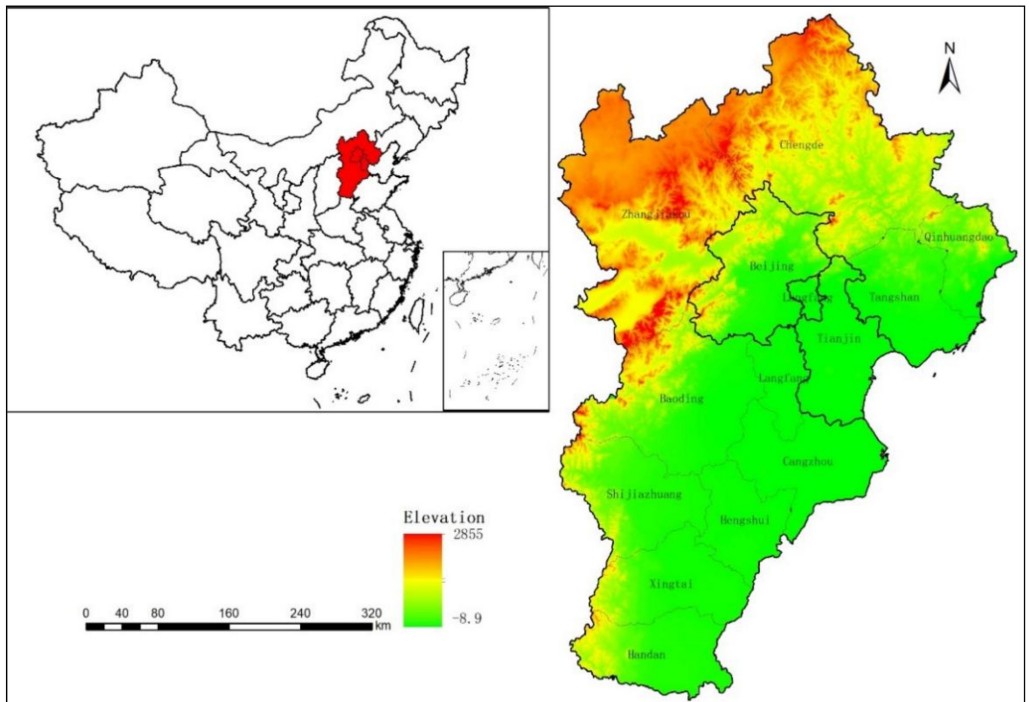

**Figure 1.** Map showing the location of the study area.

*2.2. Data Collection*

The LULC datasets (1990, 2000, 2010, 2020) were obtained from the Resource and Environment Data Center of the Chinese Academy of Sciences (http://www.Resdc.cn Accessed on 11 December 2021); the dataset was generated by manual visual interpretation using Landsat TM/ETM/OLI remote sensing images of various phases as the primary data source with a spatial resolution of 30 m. The LULC types were classified into six categories: farmland, woodland, grassland, water area, constructed land, and unused land; the overall accuracy of the 2020 LULC map is 86.4%. The DEM data were obtained from NASA (https://search.asf.alaska.edu Accessed on 11 December 2021), the elevation data collected with the ALOS (Advanced Land Observing Satellite, launched in 2006) satellite phased array L-band synthetic aperture radar (PALSAR) with a spatial resolution of 12.5 m. The river, temperature, precipitation, and road data were obtained from the Hebei Ecological Local Database of the Hebei Institute of Geographical Sciences; the river spatial distribution data were extracted based on the China's 1: 1,000,000 river dataset, the temperature and precipitation data were based on the daily observational data of meteorological stations in the study area, which were generated through sorting, calculations, and spatial interpolation processing. The socioeconomic data (population, gross domestic product (GDP)) were obtained from the Hebei Economic Yearbook. These data were mainly used to research the driving factors of the land use change.

*2.3. Scoring Matrix for ES*

This study used a matrix approach [26,27] linking 21 ES indicators (on the *x*-axis) with six land use types (on the *y*-axis) to assess ES supply and demand across the study area. This method divides ecosystem services into three categories: regulation services, supply services and cultural services, among which regulation services include eight ES indicators such as climate regulation, air quality regulation, and water quantity regulation; supply services include eight ES indicators such as crop production and supply, energy supply and biological product supply, and cultural services include six services such as recreation, landscape aesthetic enjoyment, knowledge, and education. The ES indicator value of each land use type was scored from "0–5", with 0 indicating no relevant supply or demand and 5 indicating the maximum relevant supply or demand. Based on the actual situation in the Beijing-Tianjin-Hebei region, this study set June as the evaluation month. On the basis of consulting 10 land ecology experts, the scores of the matrix were revised, and the potential supply score matrix (Table 1), actual supply score matrix (Table 2) and actual demand score matrix (Table 3) of ES in the Beijing-Tianjin-Hebei region were obtained.

**Table 1.** Ecosystem service potential matrix for the Beijing-Tianjin-Hebei region.

| | Regulating services | Climatic regulation | Air quality regulation | Water flow regulation | Water purification | Erosion regulation | Natural hazard regulation | Pollination | Regulation of waste | Provisioning services | Crops | Biomass for nergy | Livestock(domestic) | Timber | Aquaculture | Freshwate | Shipping | Mineral resources | Cultural services | Recreation & tourism | Landscape aesthetics & inspiration | Knowledge systems | Culture heritage & cultural diversity | Natural heritage & natural diversity |
|---|---|---|---|---|---|---|---|---|---|---|---|---|---|---|---|---|---|---|---|---|---|---|---|---|
| Farmland | 2 | 1 | 2 | 0 | 0 | 1 | 2 | 1 | | 5 | 5 | 0 | 0 | 0 | 0 | 0 | 0 | | 1 | 1 | 2 | 3 | 0 |
| Woodland | 5 | 5 | 4 | 5 | 5 | 4 | 4 | 5 | | 0 | 1 | 0 | 5 | 0 | 0 | 0 | 0 | | 5 | 5 | 5 | 4 | 5 |
| Grassland | 2 | 1 | 1 | 3 | 5 | 1 | 1 | 1 | | 0 | 1 | 3 | 0 | 0 | 0 | 0 | 0 | | 3 | 4 | 4 | 3 | 3 |
| Water area | 4 | 0 | 5 | 3 | 0 | 3 | 0 | 5 | | 0 | 1 | 0 | 0 | 5 | 5 | 4 | 0 | | 5 | 4 | 4 | 3 | 3 |
| Constructed land | 0 | 0 | 0 | 0 | 2 | 0 | 0 | 0 | | 0 | 0 | 0 | 0 | 0 | 0 | 0 | 0 | | 3 | 3 | 2 | 1 | 0 |
| Unused land | 1 | 0 | 1 | 2 | 2 | 3 | 1 | 2 | | 0 | 0 | 0 | 0 | 0 | 0 | 0 | 0 | | 2 | 3 | 3 | 2 | 2 |

**Table 2.** Actual supply of ecosystem services in the Beijing-Tianjin-Hebei region.

| | Regulating services | Climatic regulation | Air quality regulation | Water flow regulation | Water purification | Erosion regulation | Natural hazard regulation | Pollination | Regulation of waste | Provisioning services | Crops | Biomass for nergy | Livestock(domestic) | Timber | Aquaculture | Freshwater | Shipping | Mineral resources | Cultural services | Recreation & tourism | Landscape aesthetics & inspiration | Knowledge systems | Culture heritage & cultural diversity | Natural heritage & natural diversity |
|---|---|---|---|---|---|---|---|---|---|---|---|---|---|---|---|---|---|---|---|---|---|---|---|---|
| Farmland | | 2 | 1 | 2 | 0 | 0 | 1 | 3 | 1 | | 4 | 4 | 0 | 0 | 0 | 0 | 0 | 0 | | 1 | 1 | 1 | 1 | 0 |
| Woodland | | 5 | 5 | 3 | 4 | 5 | 3 | 1 | 5 | | 0 | 1 | 0 | 1 | 0 | 0 | 0 | 0 | | 4 | 4 | 4 | 2 | 4 |
| Grassland | | 1 | 0 | 1 | 2 | 3 | 1 | 1 | 1 | | 0 | 0 | 1 | 0 | 0 | 0 | 0 | 0 | | 2 | 3 | 3 | 2 | 2 |
| Water area | | 3 | 0 | 3 | 2 | 0 | 3 | 0 | 5 | | 0 | 0 | 0 | 0 | 3 | 2 | 2 | 0 | | 5 | 4 | 3 | 2 | 2 |
| Constructed land | | 0 | 0 | 0 | 0 | 2 | 0 | 1 | 0 | | 0 | 0 | 0 | 0 | 0 | 0 | 0 | 0 | | 2 | 2 | 2 | 1 | 0 |
| Unused land | | 1 | 0 | 1 | 1 | 1 | 2 | 1 | 1 | | 0 | 0 | 1 | 0 | 0 | 0 | 0 | 0 | | 1 | 2 | 2 | 1 | 1 |

**Table 3.** Demand for ecosystem services in the Beijing-Tianjin-Hebei region.

| | Regulating services | Climatic regulation | Air quality regulation | Water flow regulation | Water purification | Erosion regulation | Natural hazard regulation | Pollination | Regulation of waste | Provisioning services | Crops | Biomass for nergy | Livestock(domestic) | Timber | Aquaculture | Freshwater | Shipping | Mineral resources | Cultural services | Recreation & tourism | Landscape aesthetics & inspiration | Knowledge systems | Culture heritage & cultural diversity | Natural heritage & natural diversity |
|---|---|---|---|---|---|---|---|---|---|---|---|---|---|---|---|---|---|---|---|---|---|---|---|---|
| Farmland | | 2 | 1 | 2 | 0 | 3 | 2 | 3 | 2 | | 0 | 1 | 0 | 0 | 0 | 0 | 0 | 0 | | 0 | 0 | 1 | 1 | 0 |
| Woodland | | 0 | 0 | 0 | 0 | 0 | 0 | 0 | 0 | | 0 | 0 | 0 | 0 | 0 | 0 | 0 | 0 | | 0 | 0 | 0 | 0 | 0 |
| Grassland | | 1 | 0 | 0 | 0 | 1 | 1 | 1 | 1 | | 0 | 0 | 0 | 0 | 0 | 0 | 0 | 0 | | 0 | 0 | 0 | 0 | 0 |
| Water area | | 0 | 0 | 0 | 0 | 0 | 0 | 0 | 0 | | 0 | 0 | 0 | 0 | 0 | 0 | 0 | 0 | | 0 | 0 | 0 | 0 | 0 |
| Constructed land | | 5 | 5 | 4 | 5 | 2 | 5 | 1 | 3 | | 5 | 5 | 5 | 3 | 5 | 5 | 5 | 4 | | 4 | 4 | 3 | 4 | 4 |
| Unused land | | 0 | 0 | 0 | 0 | 0 | 0 | 0 | 0 | | 0 | 0 | 0 | 0 | 0 | 0 | 0 | 0 | | 0 | 0 | 0 | 0 | 0 |

*2.4. Ecological Indexes of the Relationships between Supply and Demand*

Two potentially relevant indicators for comparisons between regions are the ecosystem service supply rate and the supply/demand ratio [3]. The supply rate is used to reflect the ability of ecosystem to provide actual ecosystem services, and the supply/demand ratio is used to reflect the balance between actual supply of ecosystem and human demand. The calculation formulas are:

$$X/Y/Z = \sum_{i=1}^{n}\sum_{j=i}^{m}(A_i \bullet p_{ij}), \tag{1}$$

$$C_1 = \frac{Y}{X} \tag{2}$$

$$C_2 = \frac{Y-Z}{(X+Y)/2} \left\{ \begin{array}{ll} > 0, & surplus \\ = 0, & balance \\ < 0, & deficit \end{array} \right\} \tag{3}$$

where $X$, $Y$, and $Z$ are the ES potential supply, ES actual supply, and ES demand, $A_i$ is the area of land use type $i$, and $P_{ij}$ is the score of the potential supply, actual supply, and actual demand of ecosystem service type $j$. $C_1$ and $C_2$ are the supply rate and the supply/demand ratio.

*2.5. CLUE-S Model*

The CLUE-S model consists of four input modules: the LUCC restricted area, land use type conversion rules, land demand, and spatial characteristics, and one spatial allocation module. The rule for land use type conversion determines the time power of simulation, which includes two parts: the elasticity of land use type transfer and the order of land use type transfer. Land restricted areas refer to areas that do not change, such as nature reserves and protected essential farmland areas, which can be input into the model in the form of independent layers. Land demand is calculated or estimated by external models to limit the change of each land use type in the simulation process. Spatial characteristics refer to the spatial distribution probability of each land use type, which is mainly driven by the factors affecting its spatial distribution. Spatial allocation is a process of spatial allocation of land use demand based on the analysis of land use conversion rules, restricted areas of land use, spatial distribution probability of land use and land use type map of the base year, according to the total probability. The study unit of the CLUE-S model is a grid; the grid size in this study was 2 km $\times$ 2 km.

2.5.1. Selection of Driving Factors

Understanding the drivers of land use change and how policies can alter it will be critical to meeting the challenge of maintaining ecosystem functions and biodiversity that underpin their sustainable supply [43]. The driving factors selected need to have a relatively stable impact on the land use change in the study area over a short period, and even if the impact changes, it should be a step-like change rather than a gradual one. Biophysical factors, such as hydrology, soil, geology, and landforms, and connectivity to socio-economic factors, such as transportation networks, population centers, job centers, and quality-of-life amenities were used to simulate LULC in urban areas [44]. Some research showed that ecosystem services values (ESV) were an essential driver of land use and cover changes related to urbanization [45]. The selection of driving factors should take into account the natural, social, and economic factors, as well as the availability and quantification of the data. In this study, 10 driving factors were selected, including the elevation, slope, temperature, precipitation, distance to a river, distance to the county government, distance to Beijing, population density, GDP, and ecosystem service value. CLUE-S model used binary Logistic stepwise regression to calculate the probability of each land type appearing in each pixel in the region, and made spatial allocation by comparing the occurrence probability of various land types in the same position, the calculating equation is:

$$\text{Logit}P = \ln\left(\frac{P_i}{1 - P_i}\right) = \beta_0 + \beta_1 X_{1,i} + \beta_2 X_{2,i} + \cdots + \beta_n X_{n,i}$$

where: $P_i$ is the probability of a certain LULC type $i$ in a grid cell, $X_{1,i}{\sim}X_{n,i}$ are the driving factors of LULC type $i$, $\beta_0$ is the constant, $\beta_1{\sim}\beta_n$ are the correlation coefficients of each driving factor.

In this study, redundancy analysis was chosen to diagnose the multicollinearity of driving factors. The receiver operating characteristic (ROC) was used to evaluate the accuracy of regression analysis results. The results showed that there was no multicollinearity among the 10 driving factors. All the areas under the ROC curve were greater than 0.7, indicating that the selected factor had good explanatory power. Table 4 showed the regression coefficient (β) of each LULC type and driving factors.

**Table 4.** Regression coefficient (β) of each LULC type and driving factors.

| | Farmland | Woodland | Grassland | Water Area | Construction Land | Unused Land |
|---|---|---|---|---|---|---|
| Elevation | −0.452 | 0.258 | 0.232 | 0.006 | −0.561 | 0.129 |
| Slope | −0.343 | 0.238 | 0.417 | 0.005 | −0.654 | 0.008 |
| Temperature | 0.011 | 0.007 | 0.007 | | | |
| Precipitation | 0.009 | 0.138 | 0.211 | 0.001 | −0.001 | 0.008 |
| The distance to a river | 0.015 | 0.019 | 0.021 | 0.665 | 0.012 | |
| The distance to the county government | −0.002 | −0.025 | −0.034 | | 1.352 | −0.157 |
| The distance to the center of Beijing | −0.053 | −0.001 | −0.026 | | 0.078 | −0.011 |
| Population density | −0.035 | | | | 0.093 | |
| GDP | −0.043 | | | | 0.036 | |
| Ecosystem service value | | 1.329 | 0.072 | 1.056 | −2.534 | |
| Constant | −0.072 | −1.056 | −0.014 | −2.321 | −2.568 | −0.051 |

2.5.2. Simulation Effect and Kappa Test

The spatial pattern of the land use in 2020 was obtained by inputting the current land use situation in 2010 into the CLUE-S model and setting the parameters. The kappa coefficient was used to test the simulation effect of the model. The kappa coefficient can reflect the consistency between two images. The expression is

$$\text{Kappa} = (P_o - P_c)/(P_p - P_c),$$

where Po is the proportion of congruence between the two images; $P_c$ is the proportion of expected consistency in a random situation; and $P_p$ is the percentage of consistency in an ideal situation. Six types of land use were considered in this study, so the proportion of expected consistency is 1/6 in the random case and 1 in the rational case. The kappa coefficient was calculated to be 0.877 for this study.

**3. Results**

*3.1. Spatial-Temporal LULC Changes from 1990 to 2020*

As shown in Figure 2, the main types of land use in the Beijing-Tianjin-Hebei region were farmland and forest woodland, accounting for more than 67% from 1990 to 2020. The proportion of farmland decreased from 51.79% to 46.11%, and the proportion of woodland increased from 20.99% to 21.34%. The grassland and forests were mainly distributed in the Anshan Mountains in the north and the Taihang Mountains in the west. The farmland was mainly distributed on the plains in the southeast. The constructed land was mainly distributed in the urban area. The water bodies were mainly distributed in the Bohai Bay area, and the unused land was mainly distributed on the Bashang Plateau in the northwest and the Qinhuangdao area in the east. In terms of the spatial changes, the most apparent spatial change was in the constructed land, and the growth was mainly concentrated in the areas surrounding Beijing and Tianjin. There was a large gap between the growth area of the constructed land in the study area.

As can be seen from Figure 3, in Beijing, the composition of the land use types was as follows: farmland > forest > grassland > constructed land > water bodies > unused land. In Tianjin, the composition was as follows: farmland > constructed land > water bodies > forest > grassland > unused land. In Hebei the composition was as follows: farmland > forest > grassland > constructed land > water bodies > unused land. Overall, during 1990–2020, the proportions of farmland, grassland, and unused land decreased, while water bodies, forest, and constructed land increased. It should be noted that the farmland decreased the most, by about 6%, and the construction land increased by about 6%. The other LUCC type exhibited slightly fluctuating trends. It can be seen that the increase in the construction land in the Beijing-Tianjin-Hebei region was mainly at the expense of the farmland. Table 5 showed the transition of LULC change between 1990–2020; the land use transformation was mainly from farmland to construction land. Woodland was the most

stable land type, 94.94% had not changed, while unused land was the most unstable land type, and only 34.98% had not changed.

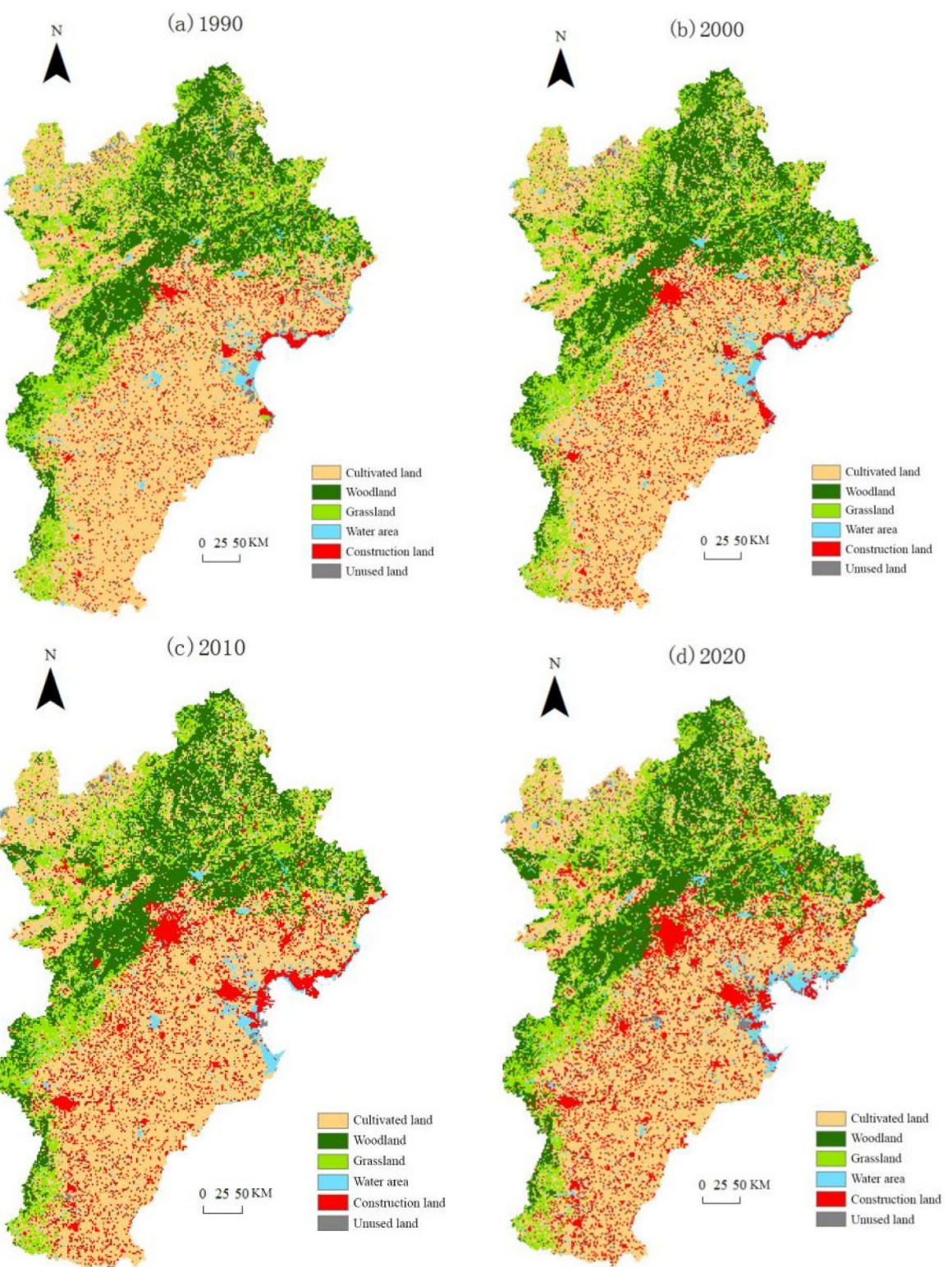

**Figure 2.** LUCC in the Beijing-Tianjin-Hebei region from 1990 to 2020.

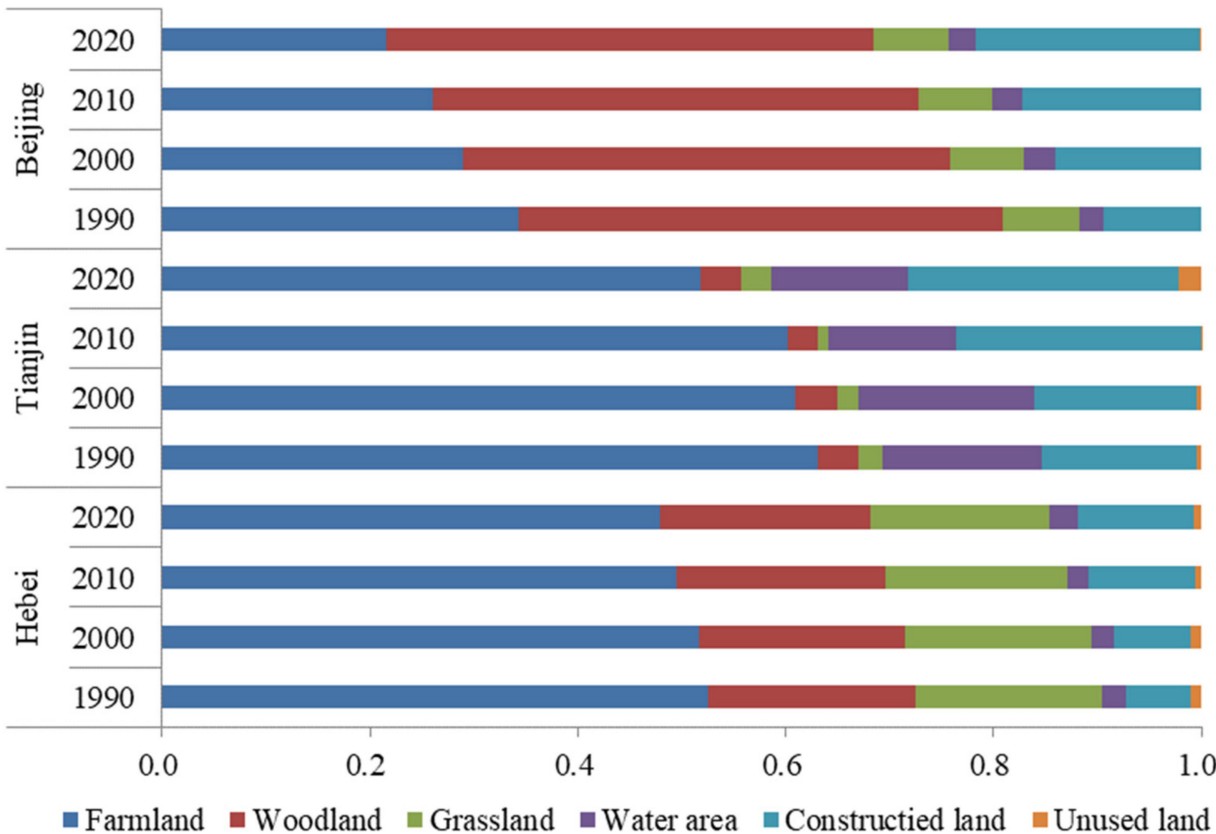

**Figure 3.** Composition of the LUCC in the Beijing-Tianjin-Hebei region from 1990 to 2020.

**Table 5.** The transition table of LULC change between 1990–2020 in the Beijing-Tianjin-Hebei region (hm$^2$).

| | | 2020 | | | | | |
|---|---|---|---|---|---|---|---|
| | | Farmland | Woodland | Grassland | Water area | Construction land | Unused land |
| 1990 | Farmland | 94612 | 1084 | 1176 | 1376 | 13408 | 104 |
| | Woodland | 552 | 42856 | 1088 | 64 | 568 | 12 |
| | Grassland | 956 | 1760 | 31004 | 368 | 1020 | 176 |
| | Water area | 800 | 116 | 264 | 4156 | 584 | 492 |
| | Construction land | 1948 | 44 | 128 | 888 | 11940 | 32 |
| | Unused land | 620 | 40 | 292 | 216 | 184 | 760 |

*3.2. Supply and Demand Patterns of ESs from 1990 to 2020*

The supply rate of the ES in the Beijing-Tianjin-Hebei region increased from 0.78 to 0.81 from 1990 to 2020. As shown in Figure 4, the supply rate of the ES in the Beijing-Tianjin-Hebei was between 0.7 and 0.9 from 1990 to 2020. The supply rate in the northern and western parts of the Beijing-Tianjin-Hebei was lower than that in the central and eastern parts, especially in the counties of Beijing and Tianjin. From the perspective of the spatial changes, the main areas around the cities changed significantly. Among them, the ecosystem service supply rate in the areas surrounding Beijing, Tianjin, Shijiazhuang, and other cities increased significantly. Compared with 1990, in 2020, the potential of the ESs in the counties with regional supply rates of 0.8–0.85 in the Beijing-Tianjin-Hebei region was further exploited and the restricted supply rates gradually increased to 0.85–0.9. The number of counties with supply rates of 0.8–0.85 decreased from 110 in 1990 to 59 in 2020, while the number of counties with values of 0.85–0.9 increased from 43 to 95. At present, the potential of the ESs is small, and the pressure of the ES supply is enormous.

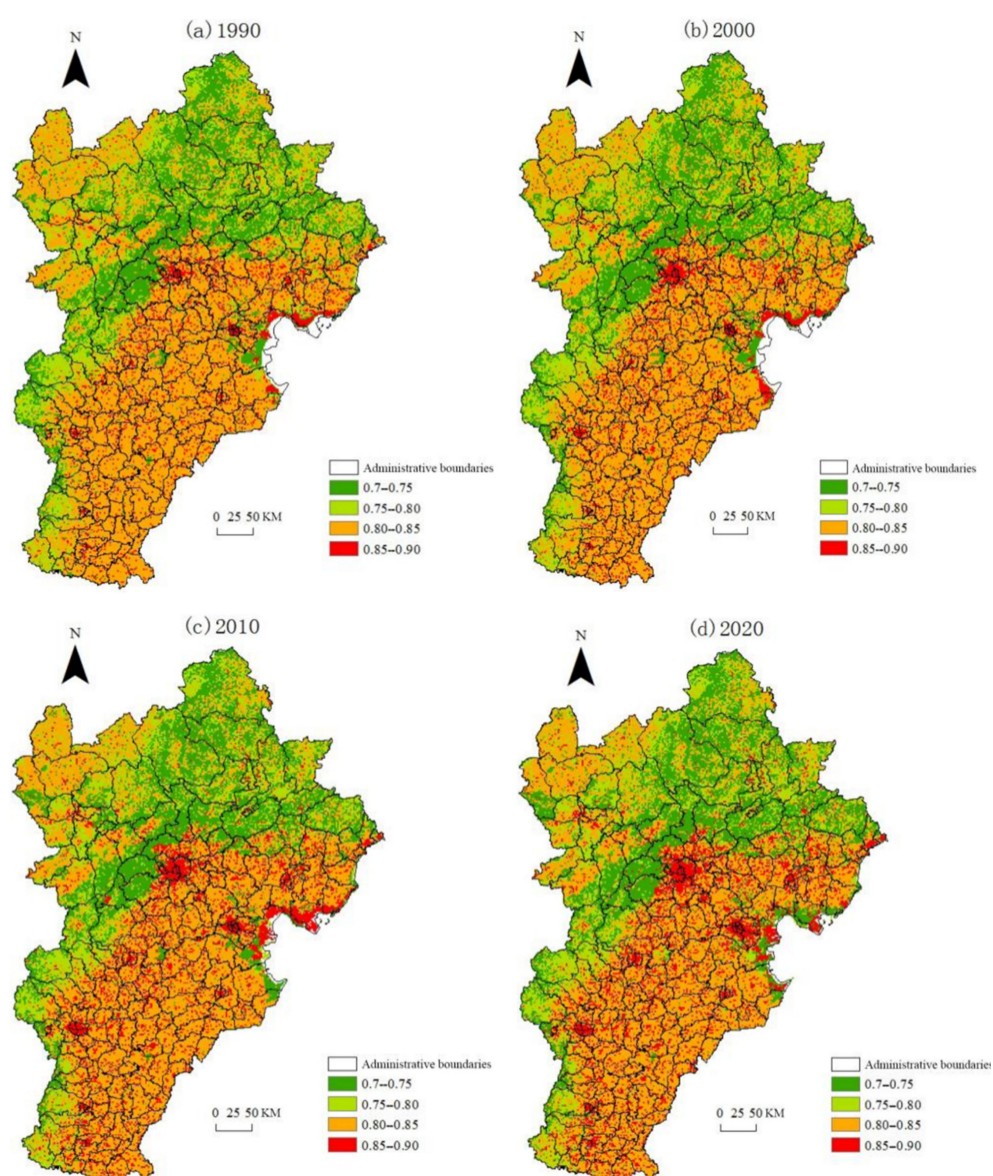

**Figure 4.** Supply rate of ESs in the Beijing-Tianjin-Hebei region from 1990 to 2020.

The supply/demand ratio of the ES in the Beijing-Tianjin-Hebei region decreased from 0.33 to 0.16 from 1990 to 2020. As is shown in Figure 5, from 1990 to 2020, the overall supply and demand pattern of the ES in the Beijing-Tianjin-Hebei region was in surplus in the northwest, it was nearly balanced in the central and southeastern areas, and cities such as Beijing, Tianjin, and Shijiazhuang were running deficits. Compared with 1990, in 2020, the supply and demand of the ESs in the northwestern part of the Beijing-Tianjin-Hebei region were quite similar, and the boundaries of the cities (e.g., Beijing, Tianjin, and Shijiazhuang) expanded rapidly. The scope of the deficit in the supply and demand of ESs increased dramatically. The number of counties in deficit increased from 144 to 168, indicating a deterioration in the supply and demand of ESs in the Beijing-Tianjin-Hebei region.

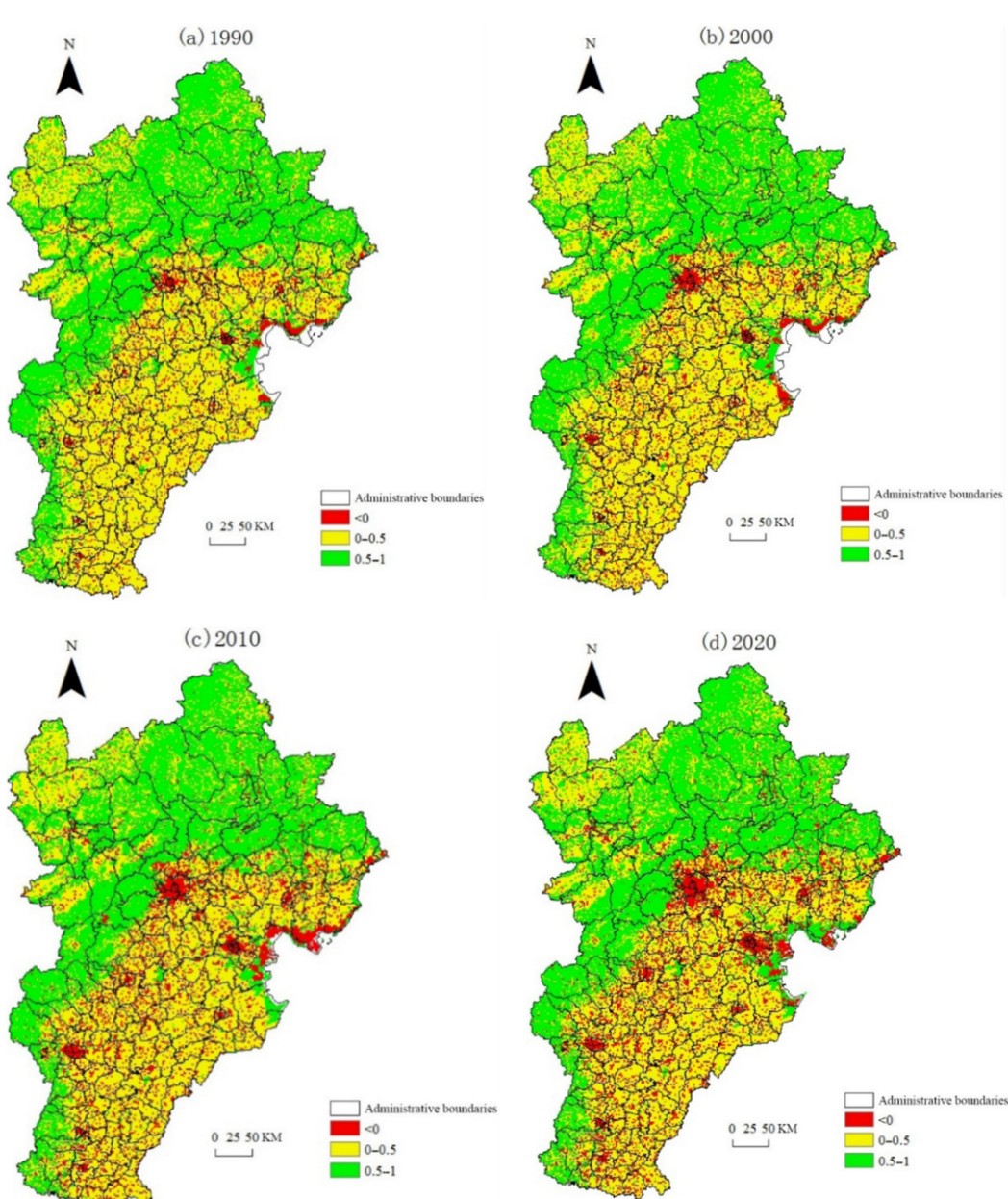

**Figure 5.** Supply/demand ratio of ESs in the Beijing-Tianjin-Hebei region from 1990 to 2020.

### *3.3. Simulation of LUCC with Multiple Objectives from 2020 to 2050*

3.3.1. Prediction of Land Use Demand with Multiple Objectives

Based on the 1990–2020 land use data for the study area, the short-term (2030), medium-term (2040), and long-term (2050) land use demands in the Beijing-Tianjin-Hebei region were calculated in different scenarios. The natural development scenario is based on the land use change pattern from 1990 to 2020, and the gray prediction model (GM) (1,1) was used to predict the future land use demand. The ecological priority development scenario mainly emphasizes protecting the ecological environment, preferring to increase the area of ecological land (woodland, grassland, and water bodies) and restricting the amount of conversion of ecological land to other types of land. The economic priority development scenario focuses on improving economic benefits by appropriately relaxing restrictions on economic development and expanding the area of construction land to improve economic benefits. From the perspective of these different goals, the constraints and the degree of constraints in each scenario are different, and the parameters of the corresponding

constraints in the linear programming are separate. In this study, the area restriction conditions for woodland, grassland, water bodies, and construction land in different scenarios were formulated referring to the local government planning documents, such as Beijing land use master planning (2006–2020), Tianjin land use master planning (2006–2020), Hebei land use master planning (2006–2020), Land and Space Planning of Hebei Province (2021–2035), Tianjin Land and Space Master Plan (2021–2035), and short-term planning of Beijing's land space (2021–2025) documents. The use target optimization model is based on the Beijing-Tianjin-Hebei region's functional zone positioning to predict land use demand in the ecological priority development and economic priority development scenarios.

The land use type demand in the different development target scenarios was calculated (Figure 6). From 2030 to 2050, with the three different development goals, the demand for built-up land will increase, and the demand for farmland and unused land will continue to decrease. It should be noted that the demand for construction land varies greatly in the different scenarios. The demand for ecological priority is 32,727–41,920 hm$^2$, and the demand for economic priority is 41,584–65,540 hm$^2$. The difference is large. The demand for construction land increases, and the demand for woodland and grassland continuously increases with the ecological priority targets; while with the natural development and economic priority targets they continuously decrease. In addition, the reduction in demand with the economic priority targets is more pronounced; the demand for water bodies with natural development and ecological priority targets increases; while its decrease with the economic priority target.

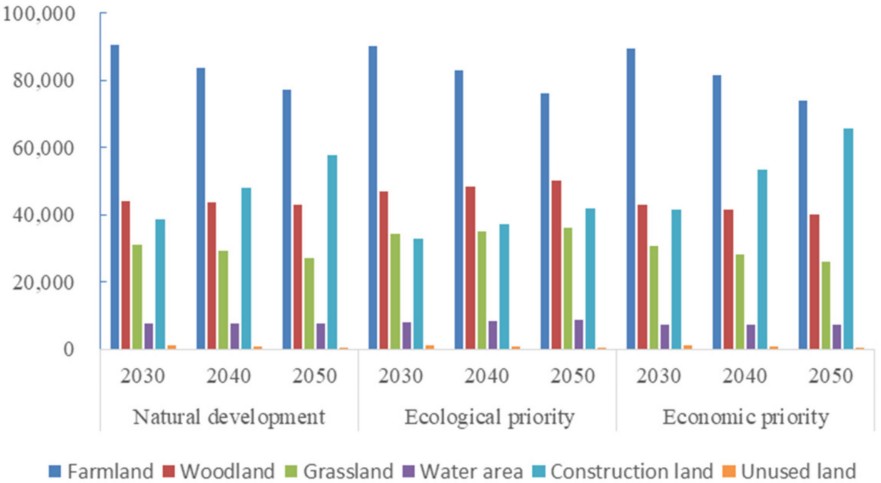

**Figure 6.** Land use demand of the Beijing-Tianjin-Hebei region with different development targets (hm$^2$).

### 3.3.2. LUCC Simulation Results with Multiple Objectives from 2020 to 2050

The coefficient of the elasticity of transition (ELAS) of land use has an essential influence on the LUCC simulation results. The ELAS measures the ease with which types can be converted from one type to another, ranging from 0 to 1. The smaller the number, the easier it is to convert the land use type. In the different scenarios, the conversion elastic coefficients are different. The results are shown in Table 6.

**Table 6.** Conversion elasticity coefficients of land use types in different scenarios.

|  | Farmland | Woodland | Grassland | Water Area | Construction Land | Unused Land |
|---|---|---|---|---|---|---|
| Natural | 0.7 | 0.8 | 0.8 | 0.8 | 0.6 | 0.3 |
| Ecological priority | 0.6 | 0.8 | 0.8 | 0.8 | 0.6 | 0.3 |
| Economic priority | 0.7 | 0.6 | 0.6 | 0.7 | 0.9 | 0.3 |

According to Figure 7, the main land use type in the Beijing-Tianjin-Hebei region from 2030 to 2050 will be farmland, which will mainly be distributed in the low plains area in Hebei Province. The forest and grassland will mainly be distributed in the Yan Mountains-Taihang Mountains and the northern part of Hebei Province. The water bodies will mainly be distributed in the eastern coastal area of Hebei Province, and the construction land will mainly be distributed in the Beijing-Tianjin area.

In the natural development scenario, the original forest land and the water bodies will remain unchanged, and the large area of construction land will expand along the original town scale. The expansion of Beijing and Tianjin will be evident, and the new construction land will be scattered within the flat area. The construction land will mainly be converted from farmland and grassland. The expansion of Beijing and Tianjin will be significant, while the Zhangjiakou and Chengde regions will experience an added sporadic distribution of new construction land. These areas will mainly be converted from farmland and grassland. Overall, the scope of farmland and grassland will decrease significantly, and the scope of construction land will increase. The area of land use with ecological benefits will be reduced, and the land use types will be spatially disordered. The regional land use will not be conducive to the protection of the ecological environment.

In the ecological priority development scenario, the areas of forest and grassland will increase in Zhangjiakou, Chengde, and Baoding. In contrast, the scope of construction land will mainly increase in Tianjin and Langfang, mainly due to the conversion of farmland. Overall, the area of land use with high ecological benefits will increase and the distribution tends will be centralized and continuous. The distribution of the construction land will be relatively optimized.

In the economic priority development scenario, the expansion of the construction land in Beijing, Tianjin, and southwestern Hebei Province will occur based on the layouts of the original cities and towns, mainly due to the conversion of farmland to construction land. New patches of construction land will expand year by year in Zhangjiakou and Chengde. The construction land in western Hebei Province will also expand, mainly due to the conversion of forest, grassland, and arable land. Overall, the layout of the construction land will be concentrated and contiguous, the forest and grassland will be scattered, the area of land with ecological benefits will significantly decrease, and the regional ecological environment situation will become worrying.

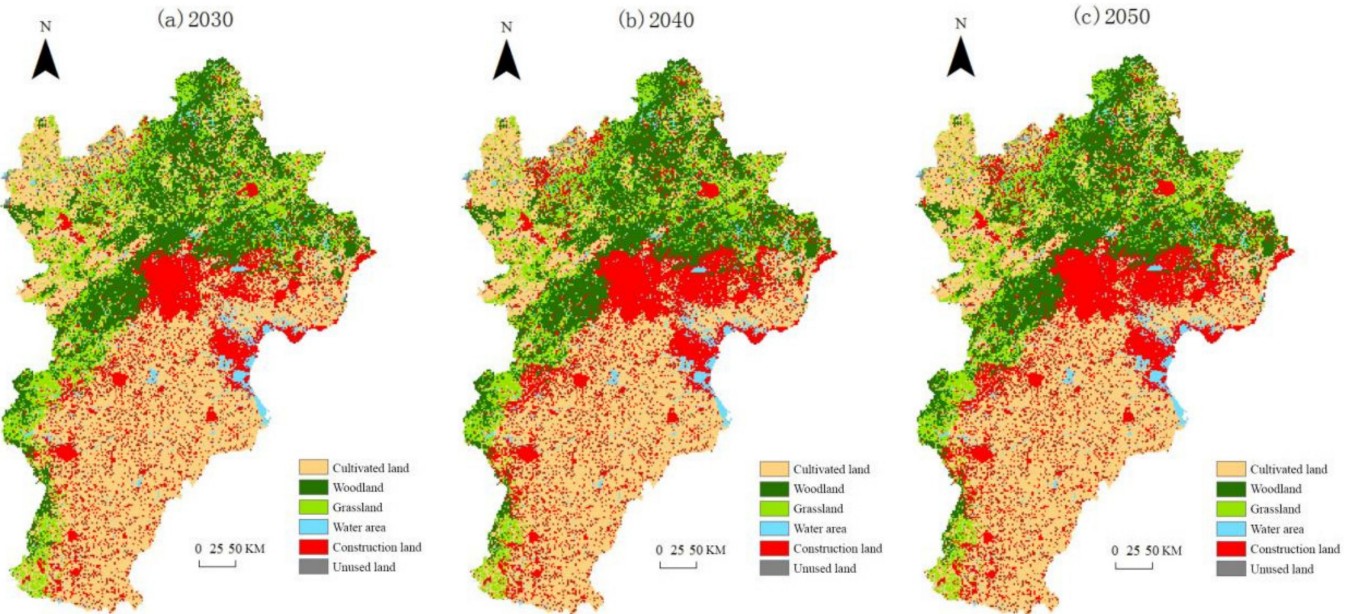

**Figure 7.** *Cont.*

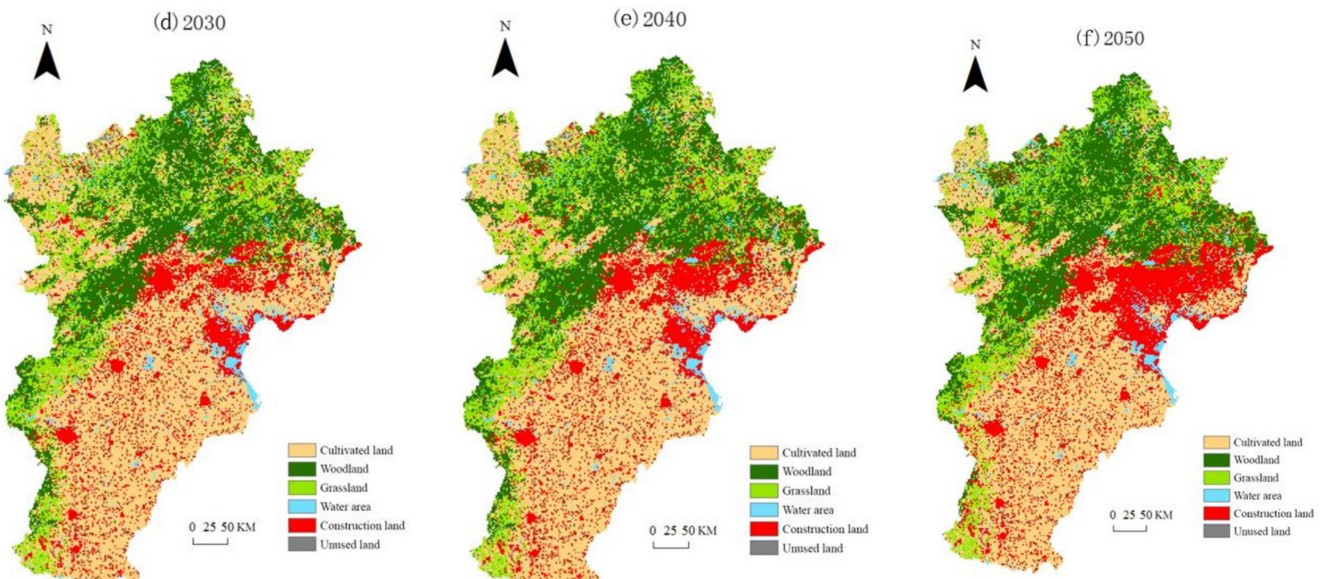

II. LUCC simulations in the ecological priority development scenario from 2020 to 2050

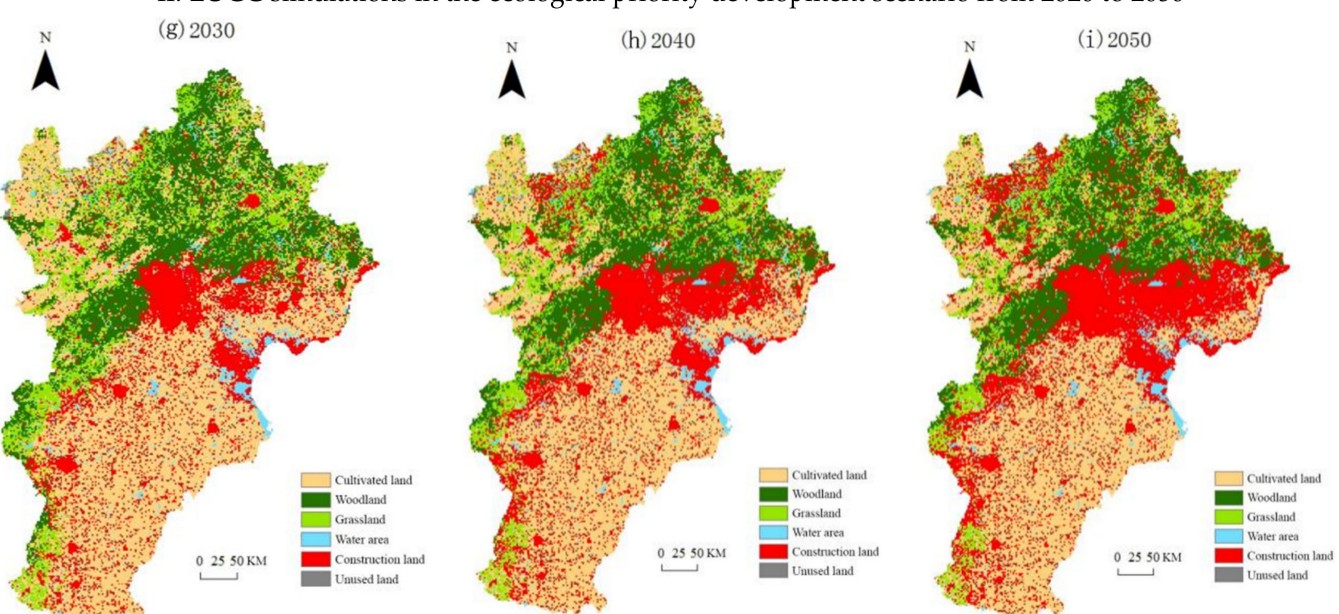

III. LUCC simulations in the economic priority development scenario from 2020 to 2050

**Figure 7.** LUCC simulation results in the Beijing-Tianjin-Hebei region for 2030–2050 under multiple objectives.

### 3.4. Analysis of Supply and Demand Pattern of ESs with Multiple Objectives

As can be seen from Figure 8, in the natural development scenario, the high value area (0.85–0.90) of the ES supply rate will be distributed in the Beijing-Tianjin area and will form a contiguous shape in the middle of the study area. The median value area (0.75–0.85) will mainly be distributed in the southeastern part of the study area, and the low value area (0.70–0.75) will mainly be distributed in the northern and western mountainous regions, forming a semi-circular shape. The spatial distribution of the ES supply rate in the ecological priority development scenario will be the same as that in the natural development scenario. The differences are mainly reflected in the Zhangjiakou area in the northwestern area and around Beijing. The supply rate values of these two areas in the ecological priority scenario will be relatively small. Compared with the natural development scenario, the high value area of the supply rate in the economic priority

development scenario will have a more extensive spatial range, mainly expanding to the eastern region; and the low value areas will become smaller, mainly in the southwest. In the natural development scenario, compared with 2020, the number of counties with values of 0.85–0.9 in 2030, 2040, and 2050 will increase to 110, 118, and 123, respectively. The potential of the ESs will be small, and the pressure of the ES supply will be enormous. In the ecological priority development scenario, the number of counties with ecosystem service supply ratios of 0.7–0.75 will increase to 7, 12, and 12 in 2030, 2040, and 2050, respectively, in the Beijing-Tianjin-Hebei region. Overall, the pressure on the ES supply will be less than in 2020. In the economic priority development scenario, the ES supply rate in the Beijing-Tianjin-Hebei region will increase from 2030 to 2050. Compared with 2020, the number of counties with values of 0.85–0.9 in 2030, 2040, and 2050 will increase to 113, 128, and 134, respectively, and the entire region will be in a high-pressure state.

As can be seen from Figure 9, in the natural development scenario, the pattern of the supply and demand of ESs in the Beijing-Tianjin-Hebei region from 2030 to 2050 exhibits a surplus in the north and a balance or deficit in the southeast. Compared with 2020, the number of counties with deficits will increase to 170, 180, and 189 in 2030, 2040, and 2050, respectively, and the number of counties with deficits will decrease to 26, 16, and 7. In the ecological priority development scenario, the counties near Tianjin will change from nearly balanced to a deficit. In contrast the counties in the northern part of the Beijing-Tianjin-Hebei region will not experience significant changes. In the economic development priority scenario, the scope of the deficit of the supply and demand of ESs will increase sharply due to the expansion of the construction land. The number of districts in deficit will increase to 171, 185, and 195 in 2030, 2040, and 2050, respectively. Compared with 2020, the number of counties close to equilibrium or in surplus will decrease year by year, and by 2050 only one county will be in surplus.

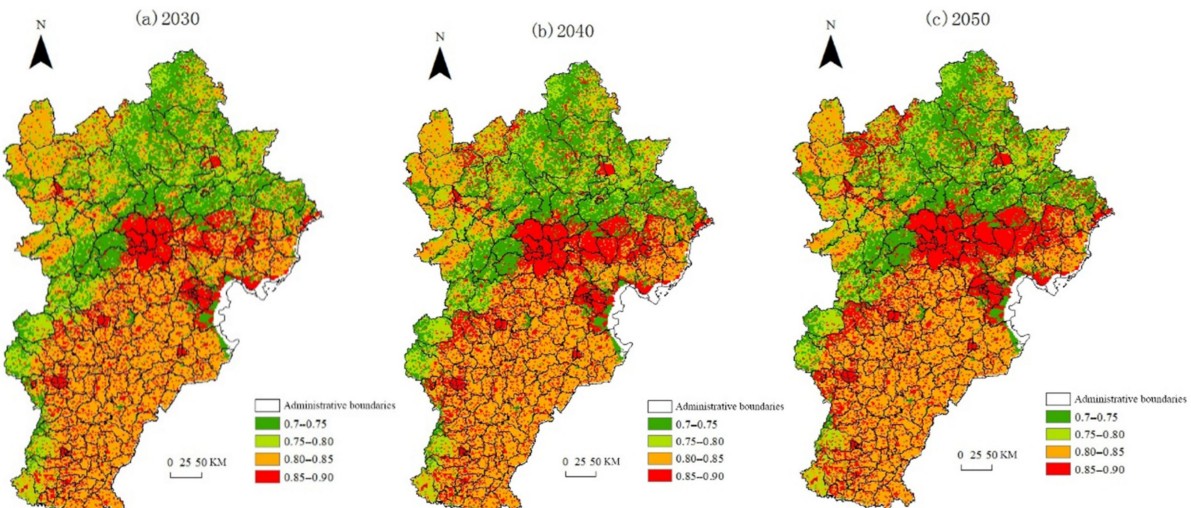

I. Supply rate distribution in the natural development scenario from 2020 to 2050

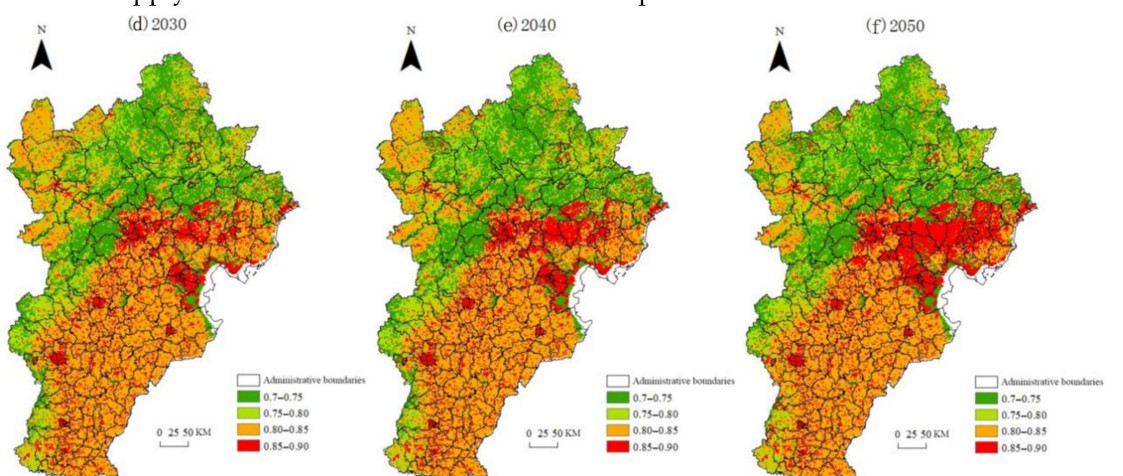

II. Supply rate distribution in the ecological priority development scenario from 2020 to 2050

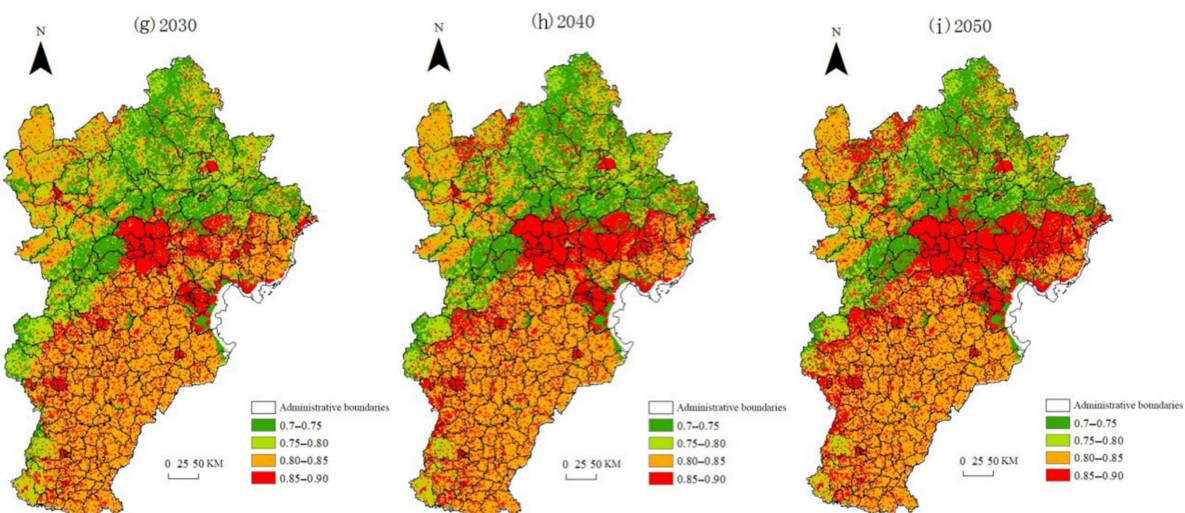

III. Supply rate distribution in the economic priority development scenario from 2020 to 2050

**Figure 8.** Supply rate distributions in the Beijing-Tianjin-Hebei region with multiple objectives from 2020 to 2050.

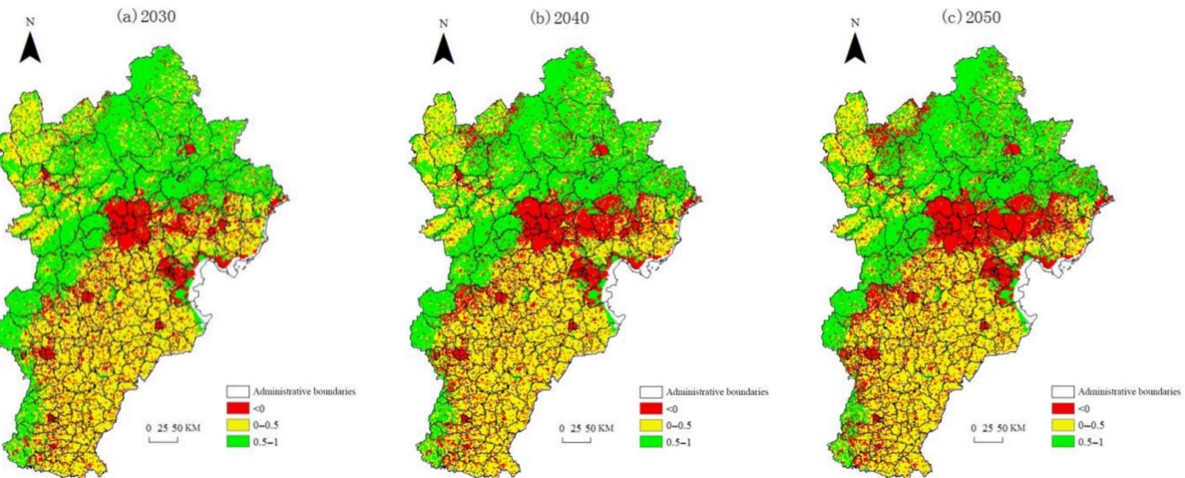

I. Supply/demand ratio distribution in the natural development scenario from 2020 to 2050

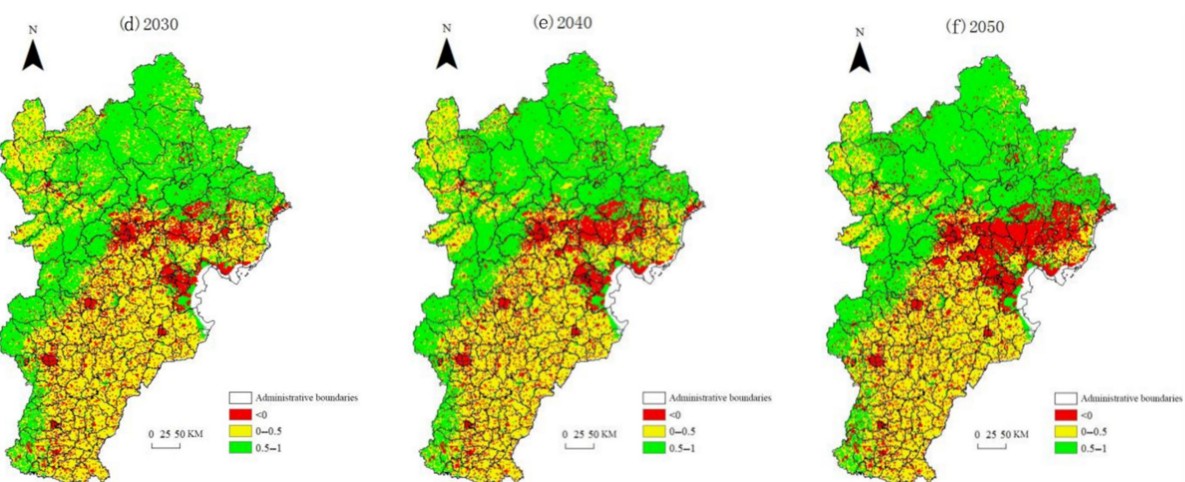

II. Supply/demand ratio distribution in the ecological priority development scenario from 2020 to 2050

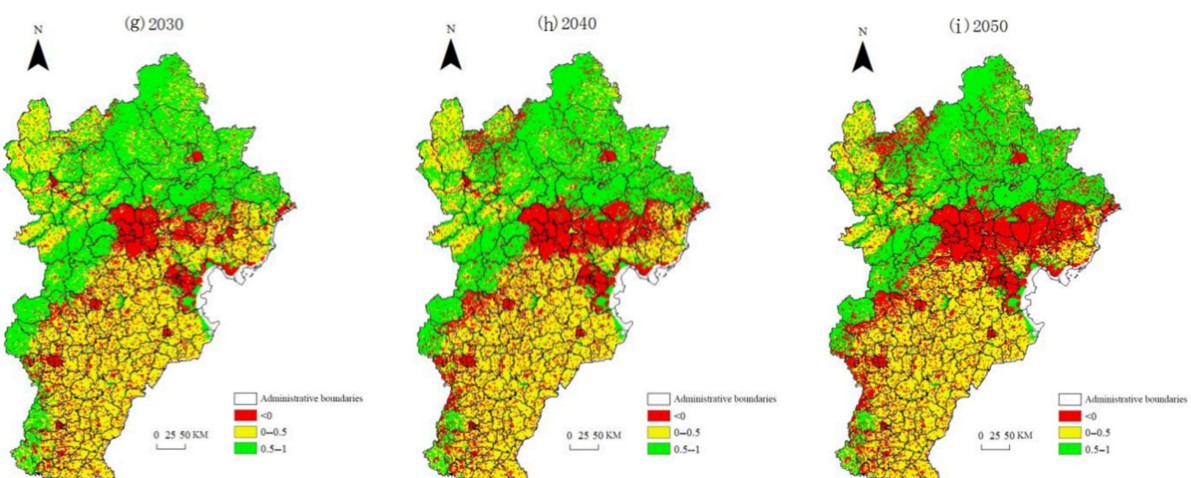

III. Supply/demand ratio distribution in the economic priority development scenario from 2020 to 2050

**Figure 9.** Supply/demand ratio distributions in the Beijing-Tianjin-Hebei region with multiple objectives from 2020 to 2050.

## 4. Discussion

In this study, based on LULC and ES score matrix, explored the spatial distribution characteristics of supply and demand of ES, and revealed the changing trend of supply and demand of ES in the Beijing-Tianjin-Hebei region from 1990 to 2020. It has important practical significance for maintaining the stability of the ecosystem in the Beijing-Tianjin-Hebei region, and promoting the sustainable development of the Beijing-Tianjin-Hebei region. The analysis of the driving factors of LULC is the basis for understanding the regular pattern of land use change and predicting future trends of land use change [46]. Elevation and slope were the main factors affecting the change of farmland and construction land, which played negative roles, so the local government should strictly control the increase of construction land in areas with high elevation and steep slopes, meanwhile, encouraged returning farmland to woodland and grassland in those areas. The distance to the county government was the most important driving factor for construction land, so the intensive use of construction land should be paid more attention. The supply and demand of ES in Bashang Plateau and Yanshan-Taihang Mountain area were in surplus, and the range of change was very tiny, showed that a series of ecological projects, such as the Beijing-Tianjin sandstorm source control project, the Three-North Shelterbelt Project, the Taihang Mountain greening project, had achieved specific results, restored the amount of local ecological land to a certain extent, and improved the regional ecosystem service function. The supply and demand of ES in plain areas of Hebei Province, Beijing and Tianjin were in deficit, the results were consistent with other scholars' related research in the same area [47–49], and the deficit was constantly increasing, especially in the areas around the cities of Beijing and Tianjin, mainly due to the rapid expansion of urban land and the increase of farmers' residential areas occupying the space of cultivated land and forest land, which leads to a sharp increase in the demand for ES. Therefore, the quantity of construction land in these areas should be strictly controlled, urban boundaries should be demarcated, and disorderly urban expansion must be strictly prohibited.

ES analysis appears to be a promising approach for a longer-term response to sustainability issues in general [50]. ES approaches have become a prominent basis for planning and management [51]. In this study, the application of the CLUE-S model, combined with the ES score matrix, suggests that the supply and demand of ES from 2030–2050 in three scenarios, the research results can provide support for the formulation of land space planning and ecological construction policies in this region. In the natural development scenario, the ES in the Beijing-Tianjin-Hebei region will remain in a high supply state from 2030 to 2050, but the pressure will be greater than before. The nature-based solutions (NBS) should be integrated in land planning [45]; place-specific strategies may include strict preservation zoning in the areas where supply and demand of ES were surplus and redlining in the areas where were deficit. The population and the amount of urban land should be strictly controlled, the red line of cultivated land protection should be delineated. The construction of ecological engineering and building ecological barriers in Yanshan-Taihang Mountain areas should be speeded up, included building soil and water conservation forests in the upper reaches of rivers, mines, reservoirs, sand sources and barren hills and wasteland suitable for forest. The protection of wetlands and farmland should be strengthened, and water-saving agriculture should be developed.

The spatial mismatch between supply and demand of ES can produce ecosystem service flow [52]. Studying the transfer path, flow direction, flow and other attributes of ES from supply area to demand area can reflect the process of ES transfer. Analyzing the quantification and spatialization of ES flow is not only helpful to identify the unsatisfied areas, regulate the process and ways of service delivery, but also helpful to the overall management of decision makers.

## 5. Conclusions

In this study, based on the current LUCC, we conducted a temporal and spatial comparative analysis of the ES supply capacity and ecosystem service supply capacity in

the Beijing-Tianjin-Hebei region. In addition, through prediction and simulation of land use changes in different scenarios, we obtained the future spatial pattern of ecosystem services. From 1990 to 2020, the supply of the ES in the Beijing-Tianjin-Hebei region was at a high level, and the ecological environment was worse. During 2030–2050, the ES of the Beijing-Tianjin-Hebei region will still be in a high supply state, and the pressure will be greater than before. The deficit, centered on the urban construction land, will widen and the ecological situation will deteriorate. In the natural development scenario, the ESs in the Beijing-Tianjin-Hebei region will continue to be in a state of high supply from 2030 to 2050, and the pressure will be greater than before. The scope of the deficit in the supply and demand of ESs, which will be centered on the urban construction land, will widen and the ecological environment around Beijing and Tianjin will deteriorate. In the ecological priority development scenario, the pressure on the ESs will be relieved, the rate of the deficit expansion will decrease, and the degree of ecological environmental deterioration around Beijing will decrease. In the economic priority development scenario, the pressure on the ES will increase sharply, the deficit area will expand rapidly, and the ecological environment around Beijing and Tianjin and in the northwestern part of the study area will deteriorate sharply. Therefore, we should balance ecology and the economy to achieve efficient land use and sustainable development in the study area.

**Author Contributions:** Conceptualization, A.W. and X.G.; methodology, A.W.; software, A.W.; validation, J.Z.; formal analysis, J.Z.; investigation, H.S.; resources, A.W.; data curation, A.W.; writing—original draft preparation, A.W.; writing—review and editing, X.G.; visualization, J.Z.; supervision, H.S.; project administration, Y.Z.; funding acquisition, A.W. All authors have read and agreed to the published version of the manuscript.

**Funding:** This study was supported by the Hebei Natural Science Foundation (No. D2018302014) and the Science and technology research project of Hebei Academy of Sciences, (China) (No. 13000022P00A64410106H).

**Institutional Review Board Statement:** Not applicable.

**Informed Consent Statement:** Not applicable.

**Data Availability Statement:** The data presented in this study are cited within the article.

**Acknowledgments:** We appreciate the constructive suggestions and comments from the editor and anonymous reviewers.

**Conflicts of Interest:** The authors declare no conflict of interest.

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
