# Peer review of "Simulation and Optimization of Supply and Demand Pattern of Multiobjective Ecosystem Services—A Case Study of the Beijing-Tianjin-Hebei Region"

_sustainability, doi:10.3390/su14052658_

Round 1

Reviewer 1 Report

The study has too many narrowly focused discussion on a locality but not enough international and theoretically relevant discussion.

The method is quite solid, but the discussion of how urban construction land has affected ecosystem services etc. etc….has been losing interests as there have been too many. TO be at least somehow interesting, this study should engage some new discussion, such as nature-based solutions.

Abstract: “It was found that the main types of land use in the Beijing- 18Tianjin-Hebei region are cultivated land and forest land”—this is not interesting enough to appear in the abstract.

Line 43:” Human beings often adopt the 43methods changing the pattern and function of land use to meet the increasing ES demand”— not understandable. Rephrase.

Line 135: “The overall accuracy of the first classification (i.e., ara- 135ble, forest, grass, water, constructed, agricultural, and unused land) is good”—instead of such generic claim, some metrics should be use.

2.5.1. Selection of driving factors—the current discussion is not satisfactory. At least some previous studies should be discussed relating to urban development, land use change and ESV:

https://www.sciencedirect.com/science/article/pii/S2213305421000205

https://www.sciencedirect.com/science/article/pii/S0013935117315748

https://esajournals.onlinelibrary.wiley.com/doi/full/10.1890/06-0750.1

https://www.sciencedirect.com/science/article/pii/S1470160X19301682

https://www.pnas.org/content/111/20/7492/

Table 1: These numbers do not offer much information. Suggest using graphs or highlight changes.

Line 394: “Although the matrix 394method is a semi-quantitative method, compared with other methods, it can evaluate the supply and demand of ecosystem services more comprehensively, and the results are 396more accurate with more evaluation indicators selected”—this is too arbitrary. I do not see how it is more accurate and comprehensive baring a comparative study.

Line 439: “The fiscal deficit, 439which will be centered on the urban construction land, will widen and the ecological en- 440vironment around Beijing and Tianjin will deteriorate.”—I do not see any “fiscal” discussion earlier in this paper.

Reviewer 2 Report

Dear authors thank you very much for your interesting research work about Simulation and optimization of supply and demand pattern of multi objective ecosystem services—A case study of the Beijing- Tianjin-Hebei region.  However, authors leave the reviewer questioning by not providing sufficient time and energy. The methodology part, result section needs extensive revision. Writing style, spelling, organization of reference requires significant editing, that clearly go beyond a major revision. 

Please check the spelling of affiliated university. I think it is Beijing Forestry University not Beijimg Forestry University.

Please include the changes percentage of LULC and ES supply / demand rate in abstract section too.

Introduction

Please review the global, regional and local level Land cover change in the beginning of the introduction section and review the other similar types of research in the similar location and present the gap analysis in details. Similarly, in this manuscript authors used CLUE-S model for the simulation however authors need to review others model (such as, CA- Markov model, LUCAS, SLEUTH, ANN, GEOMOD and provide the effectiveness of CLUE-S model in this research work. Authors may review following research work for details.

https://doi.org/10.1038/s41598-021-92299-5

 https://doi.org/10.3390/rs13204093

https://doi.org/10.3390/rs8060496

Methodology

Section 2.2 line 133: Provide the sensor and resolution details of collected LULC data of the study area with accuracy. Similarly provide the reference details also.  Rewrite this section in details and provide the references of each database.

Provide other reference work, if possible, in section 2.4 (first paragraph).

How authors provide and evaluated the driving factors of LULC change when apply in CLUE-S model, it is not clear in section 2.5. Please provide the table and its weighted of   10 driving factors selected by authors.

Section 2.5.2 Line 223:  The spatial pattern of the land use in 2020 was obtained by inputting the current land use situation in 2015 into the CLUE-S model……….. make it clear. How authors obtained current land use situation of 2015? Please further present the CLUE-S model structure in the diagram.

Result

 I don’t see the transition table of LULC change between 1990 -2020.

Section 3.2

The supply rate in the northern and western parts of Beijing-Tianjin-Hebei was lower than that in the central and eastern parts, especially in the counties of Beijing and Tianjin.  From the perspective of the spatial changes, the main areas around the cities changed significantly. Better to present this value using center point of each cities.

Section 3.3.1 Line 294 ………. formulated based on the 2006–2020 Beijing, Tianjin, and Hebei 

land use master planning documents. Please provide the clear statement and selected year 2006-2020.

Authors may discuss regression coefficient between land use type and driving factors.

Please make clear about the probability matrix of LULC simulation.

Revise the discussion section using more reference and describe the implications of this research

Reviewer 3 Report

Dear authors

Many thanks for this fascinating article. I did enjoy reading it.

While the paper presents strong results based on the case study in question, the theoretical foundation of the article needs addressing. The authors touch upon Ecosystem Services but lack to address the following:

  • The importance of green and blue infrastructure at both regional and local scales and how much of that affect land use
  • Cultural, social, economic aspects that underpin the ES theories and how that is translated in land use planning
  • Current issues, like climate change, population growth, etc. and how that is addressed through ES.

I’d encourage you to look at these articles:

  • Pedersen Zari, M. (2012). Ecosystem services analysis for the design of regenerative built environments. Building Research & Information, 40(1), 54-64.
  • Zari, M. P. (2010). Biomimetic design for climate change adaptation and mitigation. Architectural Science Review, 53(2), 172-183.
  • Marques, B., McIntosh, J., & Chanse, V. (2020). Improving Community Health and Wellbeing through Multi-Functional Green Infrastructure in Cities Undergoing Densification. Acta Horticulturae et Regiotecturae, 23(2), 101-107.
  • Renkenberger, J., Montas, H., Leisnham, P. T., Chanse, V., Shirmohammadi, A., Sadeghi, A., ... & Lansing, D. (2016). Climate change impact on critical source area identification in a Maryland watershed. Transactions of the ASABE, 59(6), 1803-1819.
  • Chan, K. M., Satterfield, T., & Goldstein, J. (2012). Rethinking ecosystem services to better address and navigate cultural values. Ecological economics, 74, 8-18.

This will allow the authors to contextualise the study results better. As it stands, there is a disconnect between theory and results. I’d also recommend that authors weave the core theoretical concepts in their discussion. We need a more robust and grounded discussion.

Reviewer 4 Report

The article presents the results of analysis and modeling of the development of ecosystem services based on secondary data. It uses the ecosystem services matrix method to evaluate changes in LULC. This is a fairly common practice today, so the article does not provide any major new findings. Model scenarios are also relatively standard and predictable.
Nevertheless, these are interesting results. However, it is important that the authors emphasize the innovative value of the findings in interpreting the results.
The article is written clearly and intelligibly. The methodology is well explained.
Formally, the graphical display of results is beneficial.
On the other hand, it would be appropriate to highlight, for example, the parts of the territory that are most subject to change or, conversely, the areas that are unchanged.
The format of references in the text needs to be adjusted
Line 78, 79 - acronyms are not explained
In Figure Six, the y-axis is described in Chinese characters

Round 2

Reviewer 1 Report

The revision shows good efforts and the manuscript can be published.

Reviewer 2 Report

Dear Authors

 Thank you very much for your revised version.